# Fluid Inclusion and Chemical Composition Characteristics of Emeralds from Rajasthan Area, India

**Li-Jie Qin** [1]**, Xiao-Yan Yu** [1,*] **and Hong-Shu Guo** [1,2]

1   School of Gemology, China University of Geosciences Beijing, 29 Xueyuan Road, Beijing 100083, China; 2009200043@cugb.edu.cn (L.-J.Q.); guo.hongshu@zijinmining.com (H.-S.G.)
2   Zijin Mining Group Gold Jewelry Co., Ltd., Xiamen 361026, China
*   Correspondence: yuxy@cugb.edu.cn

**Abstract:** Emerald is among the most valuable gems in the world. Over the past decade, its commercial value and geographic origin have been the focus of gemological and geological research. In this study, emerald samples from India were examined by UV-Vis-NIR, FTIR, Raman spectra analysis, EPMA, and LA-ICP-MS. Hexagonal three- and multi-phase inclusions are first reported in Indian emeralds, containing gas bubbles ($CO_2$ or $CO_2 + CH_4$), water or liquid mixtures of $H_2O + CO_2$, and solid phases inclusions (rounded crystals of siderite and dolomite, platelets of phlogopite, and magnesite). Mineral inclusions in Indian emeralds typically included phlogopite, quartz, talc, aragonite, and albite. The representative UV-Vis-NIR spectra show a distinct Fe absorption band, and one of the more typical characteristics of Indian emeralds is that the absorption strength of $Fe^{3+}$ (369 nm) and $Fe^{2+}$ (851 nm) is greater than that of $Cr^{3+}$ (426, 606, 635, and 680 nm). Infrared spectra show that the absorption of type II $H_2O$ is stronger than that of type I $H_2O$. LA-ICP-MS results show that Indian emeralds contain high alkali metals (10,503–16,964 ppmw; avg. 13,942 ppmw), moderate Fe (2451–4153 ppmw; avg. 3468 ppmw), low V (37–122 ppmw; avg. 90 ppmw), and the content of Cr (106–6310 ppmw) varies in a wide range. From a greenish-white core to a medium-green rim, the content of Fe, V, Cr, Sc, Cs, Rb, and Ga gradually increases in emerald with color band.

**Keywords:** Indian emeralds; fluid inclusion; spectroscopy; chemical composition



## 1. Introduction

Emerald is the gem-grade green beryl, of which chemical formula is $Be_3Al_2[SiO_3]_6$. With the discovery of new emerald deposits and the overall development and growth of the gemstone market, geographic source has become an important factor of fine-quality emerald. Consequently, studies aimed at linking each emerald to its source continue to be a central topic for geologists, gemologists, archaeologists, historians, and others [1]. Colombia, Brazil, and Zambia remain at the top of the list of major emerald producers [2]. Most notable in terms of high-quality production are the deposits of Brazil, Russia, and Zambia, but smaller deposits with high-quality emeralds have been uncovered in Madagascar and Ethiopia [3]. For the Chinese market, emeralds from Pakistan and Afghanistan have been sought after in recent years [4]. Lately, the authors discovered Rajasthan emeralds from India in a China market. Indian emeralds are typical schist-hosted emeralds [1]. As one of the most important emerald deposit types, schist-hosted emeralds mostly occur within hydrothermal quartz veins or pegmatites, which are hosted by the surrounding schists and also related to regional granitoids [5]. These granitoids provide necessary Be [6–9], while the surrounding schists is the source of chromophore (Cr and/or V) [5]. With global distribution and considerable resource prospects, the market has high expectations for schist-hosted emeralds deposits [10].

Indian emeralds have a long history. The earliest record of beryl in India can be traced back to 400 BC [11], which was most likely referring to green aquamarine [12].

Until 1943, green translucent gem-quality beryl near Kaliguman in south Rajasthan was found, identified as emerald by Crookshank (1947) [13]. However, it was a by-product of exploration of beryllium and rare metals-rich micas at that time [14]. Subsequently, systematic exploration between Udaipur and Ajmer led to the discovery of the deposits at Tikki (1945) and Gum Gurha (1951) [15]. Some small-scale open-pit mines, such as Bubani and Rajgarh, were mined, and the production reached a maximum in 1955 [14,16–18]. In 1995, emerald was also discovered embedded in the wall of a well in the village of Sankari Taluka, in the southern Indian state of Tamil Nadu. These emeralds, which are Cr-dominant, occur in a lenticular belt of mica-schists which have been traced for approximately 5 km [19].

Through the investigation of emerald deposits all over the world [5], three emerald deposits have been found in India: (I) Rajasthan (Bubani, Rajgarh, Tikki, Kaliguman, and Gaongurha); (II) Sankari Taluka; and (III) Gubaranda (Orissa state). At present, scholars have mainly studied the Rajasthan deposit, although there are a few studies of the other two deposits in India.

Previous studies on Indian emerald mainly focused on the geological characteristics, formation mechanism, petrology, and geochronology of several deposits [15,20–22]. However, there are few systematic studies on the solid and fluid inclusions, spectral characteristics, and major and trace elements. Initially, scholars briefly published on the foundation of gemological information of Indian emerald [23] and described the deposit background [12]. Later, some scholars described its mineralogical characteristics [14], and mentioned and described it several times in comparison [21,22,24–28].

By 2001, emeralds from India had begun to enter the market of Jaipur as a small but steady supply [29], and scholars believe that the potential of these reserves could be fully exploited if Indian State government could implement more liberal regulations and support. With the emergence of Indian emeralds on the market, researchers have mainly studied the causes of the formation of the Rajasthan emerald deposits and mentioned some gemological characteristics of Indian emeralds in comparative studies of origin, such as the typical "comma" fluid inclusions [30,31]. In 2020, Paul Alexander conducted an in-depth study of mineral chemistry and geochronology by selecting samples from four mines in Rajasthan.

In this paper, we collected emerald samples of Rajasthan area, India, to investigate their gemological, spectral, and chemical characteristics, so as to summarize and supplement the previous research data about Indian emeralds and provide more new research basis for the origin identification of Indian emeralds.

## 2. Geology

All known Indian Rajasthan emerald occurrences are located in the narrow and straight "Emerald Belt of India", which is oriented NE–SW and is ca. 200 km in extent [20] (Figure 1). The country rocks mainly belong to the Delhi, Aravalli, and Vindhian Groups and comprise a suite of Precambrian phyllites, biotite and muscovite schists with minor feldspar, talc schists, and vermiculite schists, associated with altered peridotites and intruded by quartz veins, granitic pegmatites, and tourmaline-bearing granitic rocks [21].

Emerald deposits in Rajasthan belong to the (fairly common) type IA, as defined by Giuliani et al. (2019) [5]: tectonic-magmatic-related type, related to pegmatites intruding mafic and ultramafic rocks. All the Rajasthan emerald deposits result from the metasomatic reaction between the pegmatites and their ultrabasic host-rocks (or Delhi gneisses as at Bubani), which is usually hosted in the contact between muscovite ($\pm$ garnet $\pm$ tourmaline) pegmatites and lenticular bodies of altered ultra-mafic rock located in the Delhi Group gneisses [15].

The best Indian emeralds come from the Rajghar deposit, found only in the phlogopite schist. The Rajghar deposit is situated ca. 20 km south of the city of Ajmer (26°17′ N, 74°38′ E; Figure 1), and the mineralizing host-rock include mainly orthogneisses and biotite schists cut by lenticular bodies of ultramafics (peridotites) and amphibolites [15].

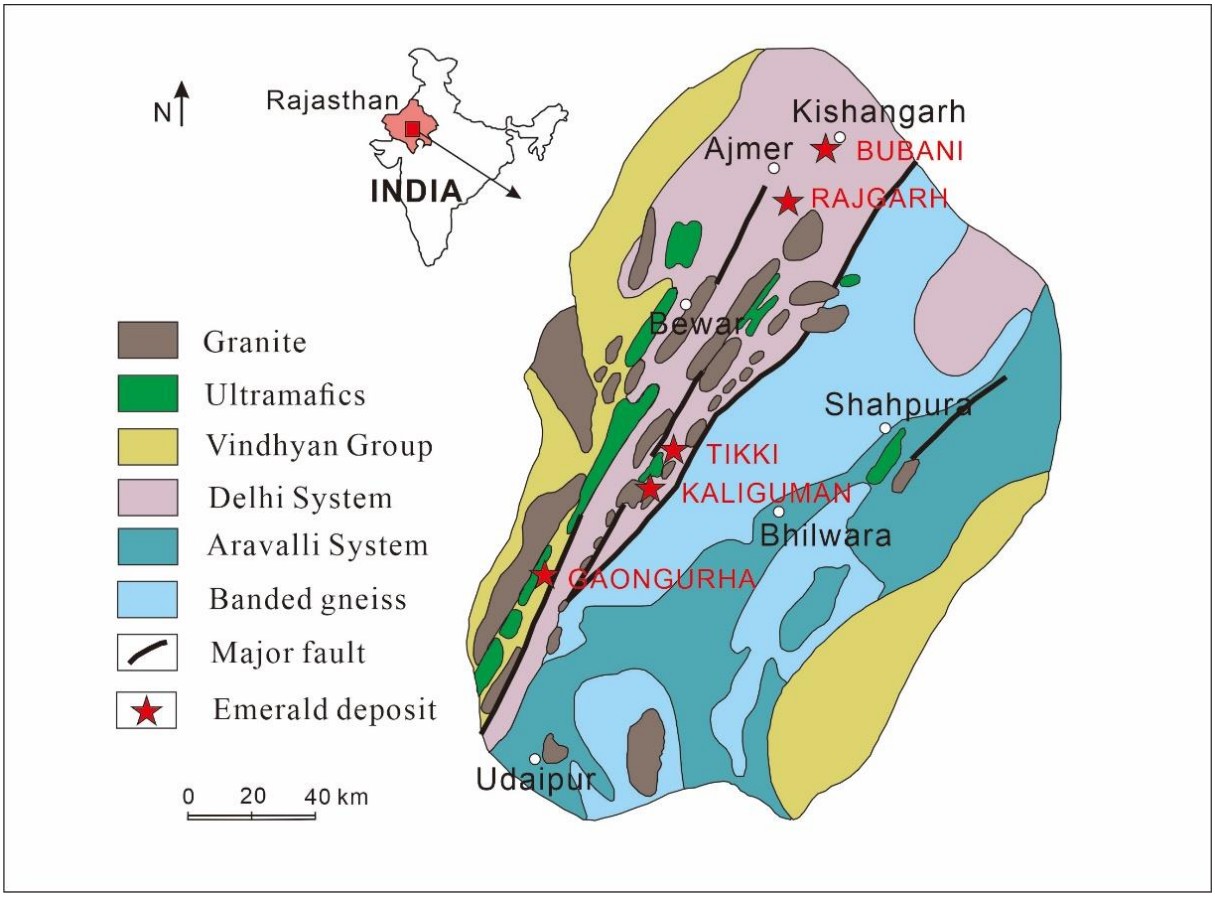

**Figure 1.** Simplified geological map of the Rajasthan Emerald Belt, NW India, situated approximately at the contact between the Delhi System, to the NW (calc-silicate rocks, limestones, sandstones, schists, and gneisses), and the Aravalli System, to the SE (mostly schists and gneisses). Modified from [15,20,32].

Indian emeralds are mostly formed in mica belts and, often, a large number of dark brown scaly mica inclusions are located parallel to the bottom of the crystals. It is also common to find rectangular hexagonal columnar megacrysts filled with gas and liquid. As the corner of the gas–liquid inclusion has a "small tail", it is called "comma" inclusion. When mica and "comma" inclusions exist at the same time, the two groups of inclusions usually intersect each other at a certain angle [30]. However, no typical "comma" inclusions were found in this study.

Paul Alexander (2020) [15] analyzed phlogopite in the phlogopite schist (host rocks of Indian emeralds) as well as potassium feldspars, plagioclase, and muscovites in pegmatite with electron microprobe to determine the chemical characteristics of the emerald deposit. The K-Ar analyses of syngenetic phlogopite and muscovite shows that the age of emerald deposition in Rajasthan is about ca. 790 ± 21 Ma [15], close to that of the last major orogeny affecting the region [27].

## 3. Materials and Methods

A total of 13 emerald crystals from India (Figure 2), with weight ranging from 2.59 to 13.31 ct and a size of 8 to 18 mm in length were analyzed. Nine emeralds were green, three were yellowish green, and one was bluish green. The most common shape of the studied crystals were hexagonal columns; vertical lines on the crystal surface were generally obvious, and the shell-like fracture and color zones were developed. Phlogopite can be observed on the surface of almost all Indian emerald samples; albite was also very common, and yellow-brown aragonite was identified on some samples (Figure 3).

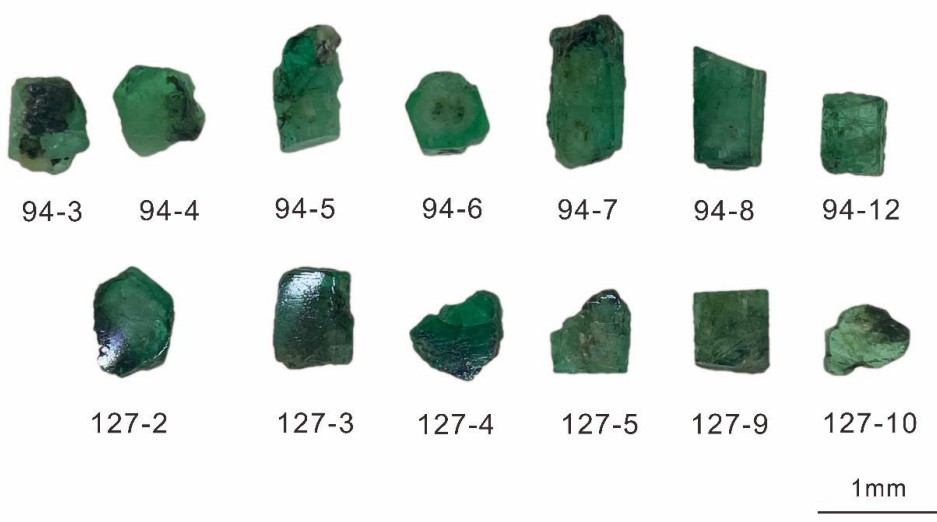

**Figure 2.** Indian emerald crystals (weight ranging from 2.59 to 13.31 ct in this photo) for this study.

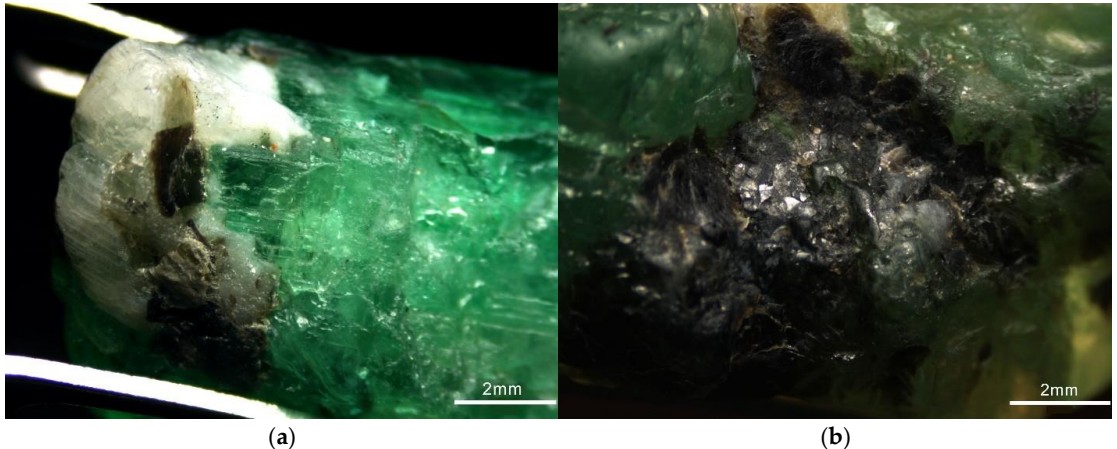

(**a**)                                             (**b**)

**Figure 3.** (**a**) Sample 94-5: White albite at one end of the surface of the hexagonal column and platelets of phlogopite; (**b**) Sample 94-4: Phlogopite and aragonite on emerald surfaces.

Standard gemological properties of all the samples were determined with a refractometer, a Chelsea filter, long-wave (365 nm) and short-wave (254 nm) UV lamps, DiamondView™, and an apparatus for hydrostatic specific gravity testing.

UV-Vis-NIR spectra were recorded on a QSPEC GEM-3000 spectrophotometer in the range of 300–1000 nm with a 0.5 nm spectral resolution and 180–200 ms integral time, housed at the gem Laboratory of the School of Gemology, China University of Geosciences in Beijing (CUGB, Beijing, China). Polarized spectra of each oriented sample were collected for obtaining ordinary ray (o-ray) and extraordinary ray (e-ray) absorption spectra.

FTIR spectra were acquired from 7200 to 2000 $cm^{-1}$ using a Bruker Tensor 27 Fourier-transform infrared spectrometer (CUGB, Beijing, China) in transmission mode and resolution set at 4 $cm^{-1}$. In the range of 4000–2000 $cm^{-1}$, each sample is tested from two directions, parallel to and perpendicular to the c-axis (∥C and ⊥C). In the range of 7200–4000 $cm^{-1}$, a clearer spectrum was obtained by testing perpendicular to the c-axis (⊥C).

Inclusions near the emerald crystal surface were identified using a Horiba LabRAM HR-Evolution Raman spectrometer housed at the gem Laboratory of the School of Gemology, China University of Geosciences in Beijing (CUGB, Beijing, China) with an Ar-ion laser operating at 532 nm excitation, between 2000 and 100 $cm^{-1}$, and accumulating up to two scans.

Internal deeper inclusions were identified using a WITec Alpha 300R confocal Raman microscopy, housed at the Innovation Academy for Earth Science, Chinese Academy of

Science, with a working wavelength of 488 nm excitation, excitation wavelength between 4000 and 100 cm$^{-1}$, actual laser power of 5.86, integration time of 1 s, and accumulating up to 64 scans.

The chemical composition of Indian emerald crystals was determined by EPMA using a JXA-IHP200F instrument housed at Key Laboratory of Metallogeny and Mineral Assessment, Institute of Mineral Resources, Chinese Academy of Geological Science (CAGS, Beijing, China). Accelerating voltage, current, and beam diameter were 10 kV, 50 nA, and 10 μm, respectively. The detection limits of individual elements ranged from 0.01% to 0.04%. Element peaks and background were measured with counting times of 10 s and 5 s, respectively. $NaAlSi_3O_8$ (NaKα, TAP), MgO (MgKα, TAP), $Al_2O_3$ (AlKα, TAP), $SiO_2$ (SiKα, TAP), $KNbO_3$ (KKα, PET), pollucite (CsLα, PET), $RbTiOPO_4$ (RbKα, PET), $CaSiO_3$ (CaKα, PET), CoO (CoKα, LIF), $MnO_2$ (MnKα, LIF), $TiO_2$ (TiKα, LIF), NiO (NiKα, LIF), $Fe_3O_4$ (FeKα, LIF), $Cr_2O_3$ (CrKα, LIF), and V-P-K glass (VKα, LIF) have been used as standards. Matrix corrections were calculated by the ZAF method [33]. The structural formulae of emerald were recalculated on the basis of 6 Si atoms per formula unit (apfu).

In situ trace elements measurements were performed using a Thermo X-Series ICP–MS fitted with a 343 nm J-100 femto-second laser ablation system Applied Spectra Inc., housed at the National Research Center for Geoanalysis, Chinese Academy of Geological Sciences (CAGS), Beijing, China. We used a laser repetition rate of 8 Hz at 1.08 J/cm$^2$ and spot diameters of ~30 μm. A baffled smoothing device was used ahead of the ICP-MS to reduce fluctuation effects induced by laser-ablation pulses and improve the quality of measurement data [34]. Each analysis consisted of ~15 s of background acquisition of gas blank measurement followed by 30 s of data acquisition from the sample. NIST SRM 610 and NIST SRM 612 were used as calibration reference materials for every 12 analyses to correct the time-dependent drift of sensitivity and mass discrimination. Data reduction was carried out with ICPMS Data Cal software and specific analytical procedures and calibration methods, and Si as internal standard element content calculated element [35].

## 4. Results

### 4.1. Gemological Properties

Most of the crystals appeared to be translucent to subtranslucent due to the relatively developed cracks, whereas crystals with fewer local cracks were transparent. Indian emeralds typically showed a green or green with yellow tone color, while several samples were bluish green. White albite and light to dark brown phlogopite were visible on emerald surfaces. Phlogopite occurred in most of the samples, extending from the surface to the interior.

Different degrees of color zones are common in Indian emeralds, and hexagonal color zones or cores were observed in the flat terminations. One sample displayed green core that turns white to light green toward the rim, and other showed a white core and a green rim (Figure 4).

The emeralds from India have a refractive index of about 1.578–1.590 ($n_e$: 1.578–1.585 and $n_o$: 1.585–1.590) with birefringence between 0.005 and 0.012. Specific gravity varied from 2.68 to 2.72, and dichroism was strong to medium, in yellowish green (o-ray) and bluish green (e-ray). Through the Chelsea filter, all the samples appeared dark green. Indian emeralds are typically inert to long- and short-wave UV radiation. According to DiamondView$^{TM}$ test, all samples showed red fluorescence with varying intensity, and some samples showed local spotted weak blue fluorescence under UV lamp, which was speculated to be the luminescence reaction of other minerals. There was also obvious disseminated yellowish brown fluorescence at crystal cracks. The visible spectrum of most emeralds from the spectroscope had distinct lines at 680 nm, partial absorption between 560 and 620 nm, and the violet range (<460 nm) was completely absorbed. The emeralds with higher transparency showed a clearer visible spectrum.

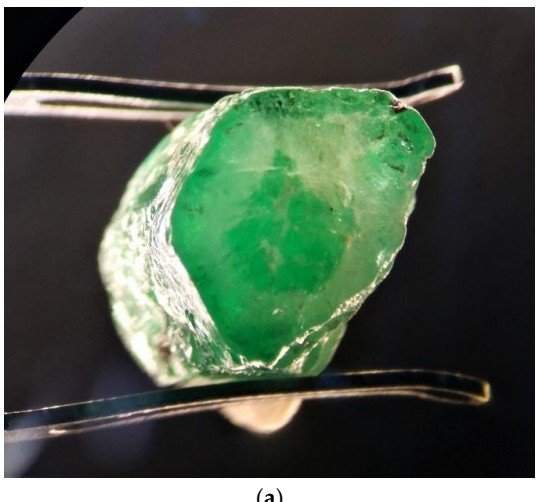
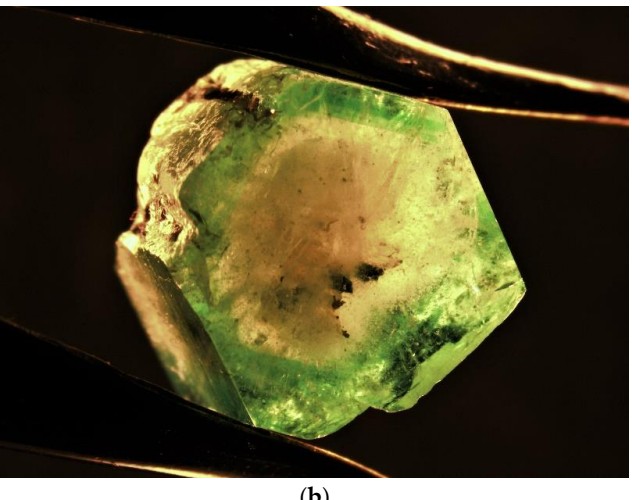

(**a**)          (**b**)

**Figure 4.** (**a**) Sample 94-5: Hexagonal color zones with green, white, and light green from the middle to the rim; (**b**) Sample 94-6: Hexagonal color zones with a white core and green at the rim. Field of view 20 mm.

The gemological properties of emeralds from India are summarized in Table 1 and discussed in detail below.

**Table 1.** Gemological properties of emeralds from India.

| Properties | Results |
|---|---|
| Color | Green or green with yellow tone, and bluish green |
| Clarity | Heavily included |
| Refractive indices | $n_e$: 1.578–1.585; $n_o$: 1.585–1.590<br>n: 1.581–1.591 [b]<br>n: 1.575–1.583 [c] |
| Birefringence | 0.005–0.012 (0.008 [c]) |
| Specific gravity | 2.68–2.72<br>2.75–2.76 [b]<br>2.73–2.75 [c] |
| Pleochroism | Strong to medium yellowish green or green (o-ray) and bluish green (e-ray) |
| Fluorescence | Typically inert to long- and short-wave UV radiation |
| Chelsea filter | Dark green |
| Visible spectrum | Distinct lines at ~680 nm; partial absorption between 560 and 620 nm; and complete absorption <460 nm |
| Internal features | <ul><li>Two-phase inclusions were typically rectangular, tubular, or of irregular shape;</li><li>Hexagonal three- and multi-phase inclusions were common in the most emeralds, containing a gas bubble and other crystals in an aqueous solution;</li><li>Hexagonal color zoning or core partially occurs on planar zones oriented perpendicular to the prism faces;</li><li>Mineral inclusions: platelets of phlogopite; quartz; talc, aragonite, and albite;</li><li>Short columnar black mineral; white cloud solid inclusion; and tubular and needle-like inclusions.</li></ul> |

All data are from the present study unless otherwise noted. [b] Data from [23], several samples from Ajmer, India. [c] Data from [29], fifteen samples from Rajasthan deposit, India.

*4.2. Magnification*

Emeralds from India contained abundant solid inclusions and fluid inclusions that can be either two-phase, three-phase, or multiphase. The most common solid inclusions in Indian emeralds were platelets of phlogopite, which were found both on the surface and inside the crystal (Figure 5a). Two black columnar mineral inclusions were observed in the greenish-white core of sample 94-6 (Figure 5b), and the EMPA and Raman test

results showed that the inclusions were quartz; the content of $SiO_2$ was 100.35 wt.%. Short columnar black mineral inclusions and white cloud solid inclusions occurred in some samples (Figure 5c,d). Indian emeralds contain a black opaque mineral and unoriented needle-shaped inclusions of light green (Figure 5e). In addition to albite on the surface, the white mineral with a misty appearance on the inside was identified as talc (Figure 5f). A group of tubular inclusions oriented parallel to the c-axis (Figure 5g) and needle-like inclusions (Figure 5h) were also visible on emerald interior.

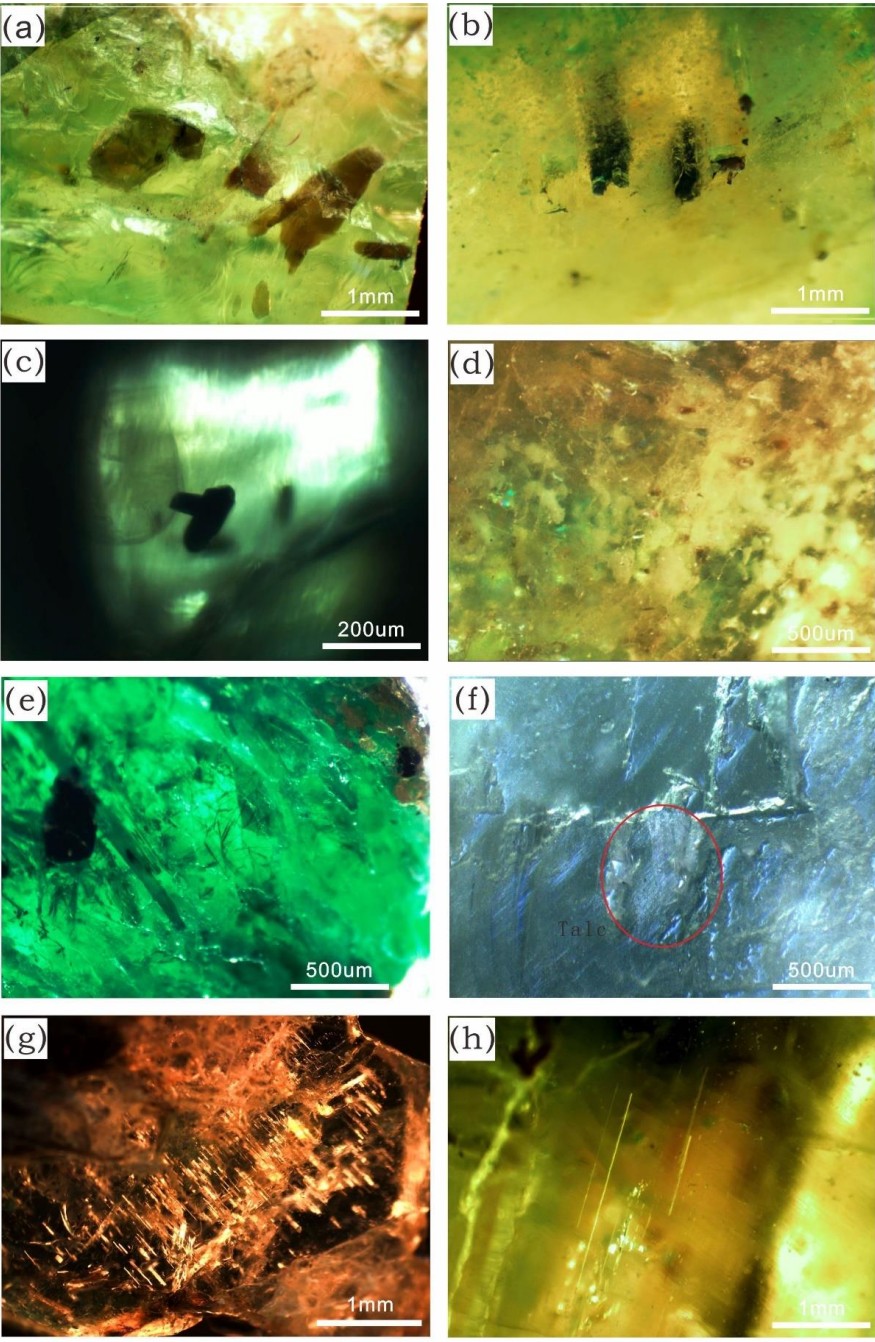

**Figure 5.** (**a**) Platelets of phlogopite were widespread in the Indian emerald samples; (**b**) Long columnar black quartz was observed in sample 94-6; (**c**) Short columnar black mineral was present in sample 94-8; (**d**) White cloud solid inclusion was observed in sample 127-9; (**e**) Sample 127-2 contains black inclusions and disorderly distributed acicular inclusions; (**f**) Talc mineral was identified in sample 94-5; (**g**) Cloud of tubes form a rain-like inclusion parallel to the c-axis in sample 94-8; and (**h**) A group of parallel needle-like inclusions were seen in sample 127-10.

Two-phase inclusions were typically rectangular, tubular, or of irregular shape (Figure 6a,b), and contained a gas bubble ($CO_2$ + $CH_4$, identified by Raman spectrometer). Rectangular fluid inclusions often appear to be composed of only two phases, a liquid and a gas. At room temperature, the gaseous bubbles in the two-phase fluid inclusions appeared to occupy one-half to two-thirds of the volume of the inclusion (Figure 6c). Groups of irregular fluid inclusion were widely distributed in the most samples (Figure 6d).

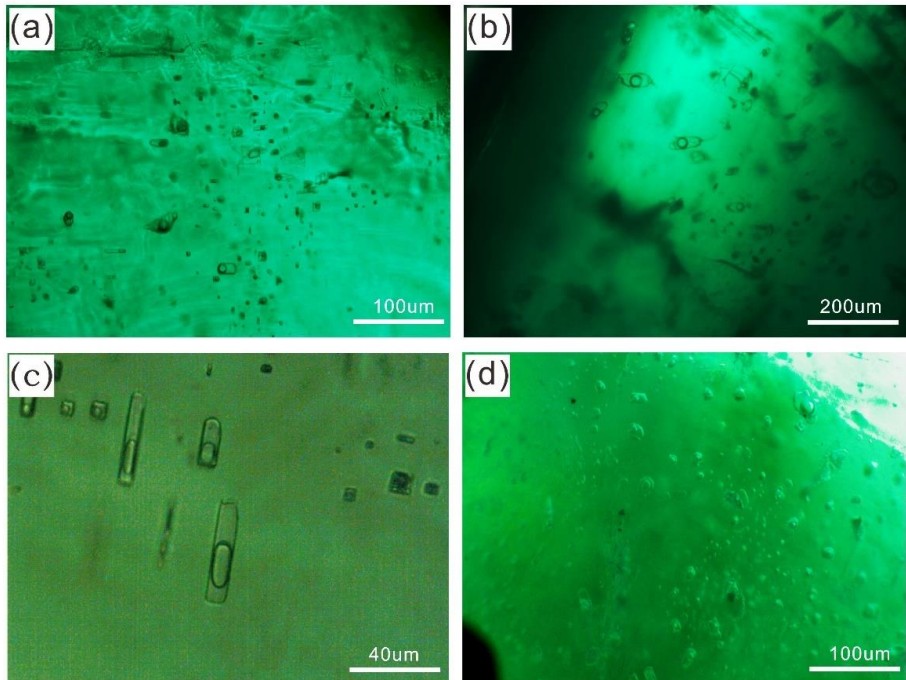

**Figure 6.** (**a**,**b**) Irregularly shaped two- and three-phase inclusions in Indian emerald; (**c**) Two-phase rectangular inclusions contained an elliptical gas bubble; and (**d**) Groups of Irregular fluid inclusion.

Rectangular two-phase inclusions, hexagonal three-phase inclusions, and irregular multiphase inclusions are common in most samples. Irregular fluid inclusions also suggest Indian emerald deposits belong to the tectonic-magmatic-related type.

### 4.3. Composition of the Multiphase Inclusions

Hexagonal fluid inclusions were common in most emeralds (Figure 7), which can be used as a characteristic of Indian emeralds. Three-phase inclusions contained a gas bubble and a crystal in an aqueous solution. Hexagonal shaped multiphase inclusion hosting a gas bubble and several crystals were observed in Indian emeralds.

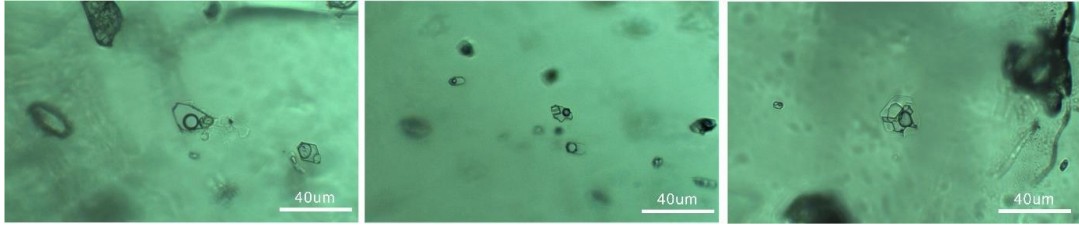

**Figure 7.** Three- and multi-phase inclusions showed a hexagonal outline in Indian emeralds.

The inclusion components of the Indian emeralds were identified by Raman analysis. The liquid phase of multi-phase inclusions was mainly water, but also included liquid mixtures of $H_2O$ + $CO_2$ ($H_2O$: sharp peaks at 3599 cm$^{-1}$; $CO_2$: peaks at 1380 cm$^{-1}$). Gas components included $CO_2$ or $CO_2$ and $CH_4$ ($CO_2$: peaks at 1282 and 1386 cm$^{-1}$; $CH_4$: sharp peaks at 2918 cm$^{-1}$). The solid phases included rounded crystals of siderite (peaks at 172,

288, 718, and very strong 1091 cm$^{-1}$; Figure 8) and dolomite (peaks at 175, 295, 721, and very strong 1094 cm$^{-1}$; Figure 9). Two distinct mineral compositions, platelets of phlogopite and magnesite (phlogopite: peaks at 3710 cm$^{-1}$; magnesite: peaks at 324 and 1092 cm$^{-1}$; Figure 10), were clearly determined in a multi-phase inclusion.

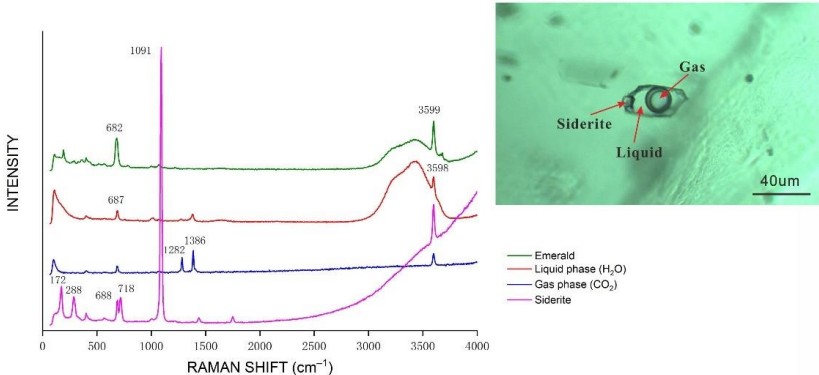

**Figure 8.** This hexagonal inclusion in Indian emerald clearly displays a gas bubble and a colorless crystal. Raman spectroscopy was used to identify the host emerald (green), the liquid phase: water (red), the $CO_2$ gas bubble (blue), and the siderite crystal (purple). The spectra are stacked for clarity.

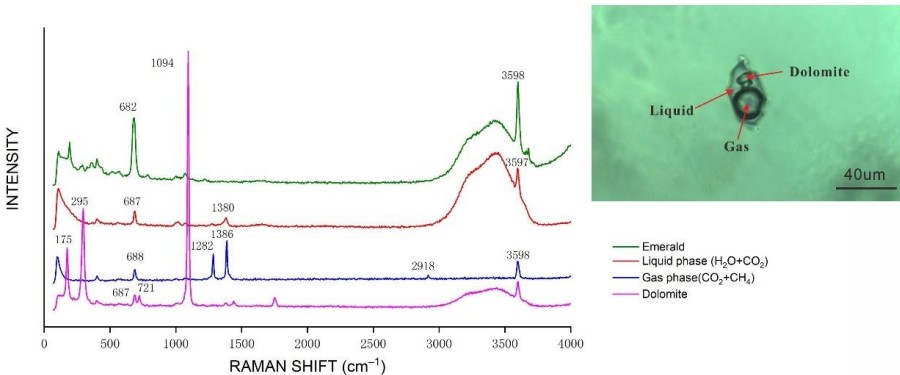

**Figure 9.** This multi-phase inclusion in Indian emerald clearly displays a gas bubble and some solid crystal inclusions. Raman spectroscopy was used to identify the host emerald (green), the liquid mixtures of $H_2O + CO_2$ phase (red), the $CO_2 + CH_4$ gas bubble (blue), and the dolomite crystal (purple). The spectra are stacked for clarity.

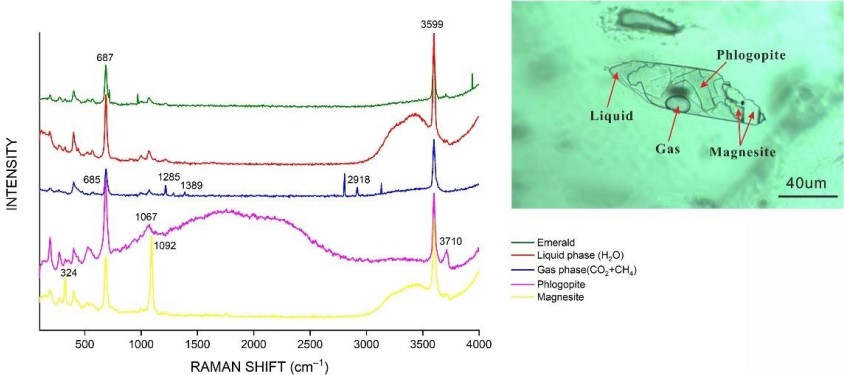

**Figure 10.** This multi-phase inclusion in Indian emerald clearly displays a gas bubble and some solid crystal inclusions. Raman spectroscopy was used to identify the host emerald (green), the liquid phase: water (red), the $CO_2 + CH_4$ gas bubble (blue), the phlogopite crystal (purple), and the magnesite crystal (yellow). The spectra are stacked for clarity.

In this study, the composition of three- and multi- phase inclusions in Indian emerald were analyzed by Raman test, which are different from the jagged three-phase fluid inclusions in Colombia emeralds. This multi-phase inclusion in Indian emerald clearly displays a gas bubble and some solid crystal inclusions. The gas phase is mainly $CO_2$ or $CO_2 + CH_4$. The solid phase is mainly carbonates (siderite, magnesite, and dolomite) and phlogopite.

### 4.4. Spectroscopy
#### 4.4.1. UV-Vis-NIR

A representative UV-Vis-NIR e-ray spectrum from the emerald host with good transparency and clarity is shown in Figure 11. The narrow band at around 369 nm indicated the presence of $Fe^{3+}$. The most prominent features in this group were the $Cr^{3+}$ bands at 426, 606, 635, and 680 nm. The wide absorption band at 851 nm is caused by different lattice sites occupied by $Fe^{2+}$ [3].

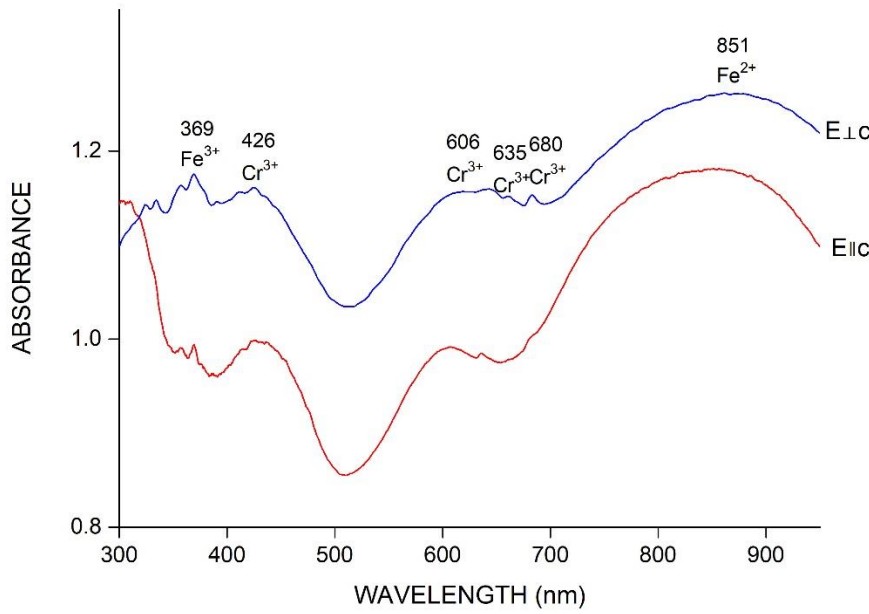

**Figure 11.** Representative UV-Vis-NIR spectra of samples 94-6 Indian emeralds.

By comparing the UV-Vis-NIR absorption spectra of the same sample in different directions, it is found that the intensity of absorption peaks caused by different chromogenic elements is different. The absorption intensities of $Fe^{3+}$ and $Fe^{2+}$ perpendicular to the c-axis were higher than those parallel to the c-axis. A distinct $Fe^{3+}$ absorption band was observed at 369 nm. Compared to Wood and Nassau (1968) [36] and Schmetzer et al. (1974) [37], the more typical characteristic of Indian emeralds is that the absorption strength of $Fe^{3+}$ and $Fe^{2+}$ is greater than that of $Cr^{3+}$.

#### 4.4.2. Infrared

Figure 12 shows the absorption peaks of Indian emerald samples in the characteristic band region (4000–2000 $cm^{-1}$) in two directions. It can be seen that the absorption in the direction perpendicular to the c-axis is more obvious than that parallel to the c-axis. The weak absorption peak near 3703 $cm^{-1}$ perpendicular to the c-axis is caused by the antisymmetric stretching vibration of type I $H_2O$, and the weak absorption peak near 3676 $cm^{-1}$ parallel to the c-axis is caused by the antisymmetric stretching vibration of type II $H_2O$ [38,39]. The absorption peaks caused by the symmetric stretching vibration of II $H_2O$ can be measured in both directions, located in the vicinity of 3600 $cm^{-1}$ ($\perp$C) and 3593 $cm^{-1}$ ($\parallel$C), respectively, and the absorption intensity is higher in the direction perpendicular to the c-axis.

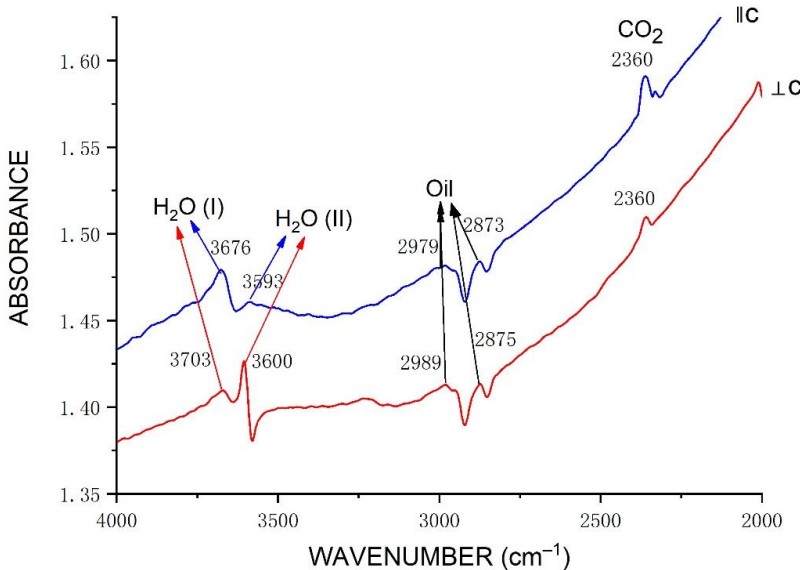

**Figure 12.** The representative FTIR spectrum of the two directions in the range of 4000–2000 cm$^{-1}$.

The significant peak at 2360 cm$^{-1}$ corresponds to $CO_2$, which was present in all the samples. The infrared spectroscopy in the 3100–2800 cm$^{-1}$ range shows 2–3 absorption peaks associated with fillers used for clarity enhancement. None of the green fillers were seen in any samples, but some of the stones containing a near-colorless filler showed infrared spectra (Figure 12) typical of cedarwood oil, as indicated by absorptions at 2873, 2875, 2979, and 2989 cm$^{-1}$ [40].

Figure 13 shows typical FTIR spectra (o-ray) in the range of 7200–4000 cm$^{-1}$. The peaks at 7078 and 5273 cm$^{-1}$ of type II $H_2O$ are sharp and strong, while the peak at 7136, 5587, and 6821 cm$^{-1}$ of type I are weak [36,41].

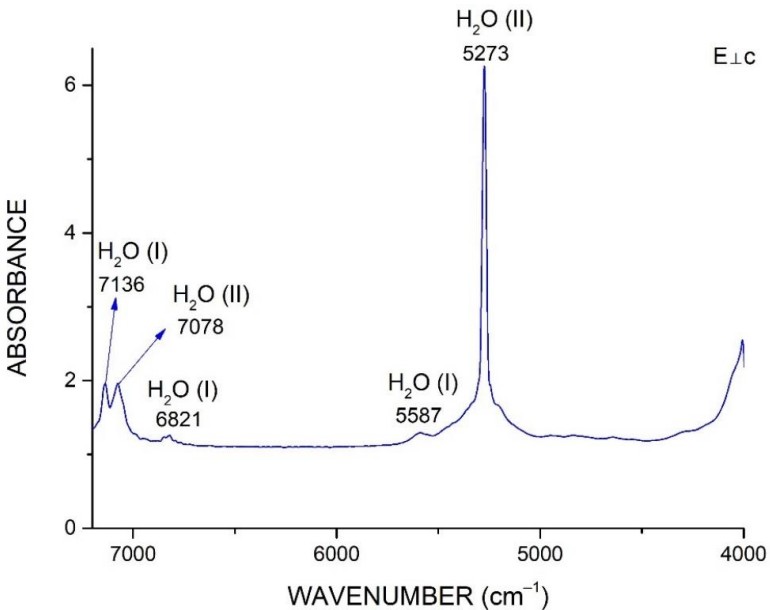

**Figure 13.** The representative FTIR spectrum in the range of 7200–4000 cm$^{-1}$ (E⊥C) illustrating type I and type II $H_2O$ observed in Indian emeralds.

### 4.5. Major and Trace Elements Analysis

Six Indian emerald samples (94-5, 94-6, 94-8, 127-2, 127-4, and 127-9) were measured by EMPA and were analyzed using LA-ICP-MS. Three different tones (yellowish green,

bluish green, and green) of emerald crystals were selected, and each sample was analyzed by 2–3 points. In order to study the cause of color zones, sample 94-6, 127-2, and 94-5 were analyzed by EMPA and LA-ICP-MS.

The results of EMPA (Table 2) from the Indian emeralds showed that the main components were $SiO_2$ (63.61–65.02 wt.%), $Al_2O_3$ (15.32–17.68 wt.%), and BeO (12.51–13.58 wt.%). The most important chromophore in Indian emeralds is Cr, which ranged from 0.01 to 0.93 wt.% $Cr_2O_3$. In contrast, $V_2O_3$ concentrations were consistently low, averaging just 0.02 wt.% and attaining a maximum of 0.09 wt.%. The other significant chromophore element was Fe, and FeO contents are between 0.38 wt.% and 0.73 wt.%. The emeralds contained relatively high contents of MgO (between 1.61 wt.% and 2.41wt.%) and somewhat less $Na_2O$ (average of 1.58 wt.%). CaO and $K_2O$ were documented in all samples, and traces of $Cs_2O$ were detected in some samples.

To compensate for the charge deficit introduced by the substitution of divalent cations for Al ions at Y site, the structure channel incorporates monovalent alkali cations [22,42,43]. $Na^+$ contents ranged from 0.219 to 0.347 apfu; $K^+$ contents ranged from 0.003 to 0.022 apfu; and $Cs^+$ contents ranged from 0.000 to 0.003 apfu.

**Table 2.** Chemical compositions (average) with the structural formulas of emeralds from India occurrence analyzed by EPMA.

| Sample | | 94-5 | 94-6 | 94-8 | 127-2 | 127-4 | 127-9 |
|---|---|---|---|---|---|---|---|
| **Oxides (wt.%)** | | | | | | | |
| $SiO_2$ | Range | 62.24–64.83 | 63.61–64.76 | 63.79–64.00 | 64.47–65.01 | 63.90–64.53 | 64.03–64.16 |
| | Average | 64.48 | 64.33 | 63.87 | 64.81 | 64.22 | 64.10 |
| $TiO_2$ | Range | bdl-0.01 | bdl-0.03 | bdl-0.04 | bdl-0.05 | bdl-0.04 | bdl-0.03 |
| | Average | bdl | 0.01 | 0.02 | 0.01 | 0.02 | 0.02 |
| $Al_2O_3$ | Range | 16.56–17.68 | 16.00–17.51 | 15.32–15.74 | 16.28–17.50 | 15.46–15.50 | 16.52–16.58 |
| | Average | 16.97 | 16.76 | 15.56 | 16.97 | 15.48 | 16.55 |
| $V_2O_3$ | Range | bdl-0.01 | bdl-0.05 | 0.02–0.05 | bdl-0.04 | 0.01–0.09 | bdl-0.01 |
| | Average | bdl | 0.02 | 0.04 | 0.02 | 0.05 | 0.01 |
| $Cr_2O_3$ | Range | 0.01–0.22 | 0.04–0.16 | 0.13–0.22 | 0.03–0.25 | 0.92–0.93 | 0.08–0.11 |
| | Average | 0.08 | 0.09 | 0.17 | 0.11 | 0.93 | 0.10 |
| BeO | Range | 12.81–13.26 | 12.73–13.58 | 13.21–13.50 | 12.51–12.92 | 13.16–13.21 | 13.34–13.36 |
| | Average | 12.99 | 13.13 | 13.31 | 12.72 | 13.19 | 13.35 |
| MgO | Range | 1.64–2.12 | 1.67–2.39 | 2.23–2.27 | 1.61–2.17 | 2.29–2.41 | 2.01–2.11 |
| | Average | 1.94 | 2.04 | 2.26 | 1.87 | 2.35 | 2.06 |
| CaO | Range | 0.03–0.05 | 0.01–0.07 | 0.03–0.05 | 0.02–0.06 | 0.04–0.04 | 0.01–0.03 |
| | Average | 0.04 | 0.04 | 0.04 | 0.04 | 0.04 | 0.02 |
| FeO | Range | 0.38–0.53 | 0.42–0.65 | 0.51–0.61 | 0.41–0.70 | 0.69–0.73 | 0.44–0.50 |
| | Average | 0.46 | 0.53 | 0.57 | 0.51 | 0.71 | 0.47 |
| $Na_2O$ | Range | 1.37–1.73 | 1.22–1.70 | 1.83–1.91 | 1.29–1.62 | 1.70–1.74 | 1.68–1.68 |
| | Average | 1.58 | 1.45 | 1.88 | 1.44 | 1.72 | 1.69 |
| $K_2O$ | Range | 0.04–0.05 | 0.02–0.07 | 0.06–0.09 | 0.05–0.18 | 0.06–0.06 | 0.04–0.04 |
| | Average | 0.04 | 0.04 | 0.07 | 0.08 | 0.06 | 0.04 |
| $Li_2O$ [a] | Range | 0.05–0.05 | 0.05–0.08 | 0.05–0.07 | 0.04–0.07 | 0.04–0.05 | 0.05–0.13 |
| | Average | 0.05 | 0.07 | 0.06 | 0.05 | 0.05 | 0.09 |
| $Cs_2O$ | Range | bdl-0.04 | 0.02–0.05 | 0.03–0.03 | 0.02–0.04 | 0.05–0.06 | 0.05–0.07 |
| | Average | 0.02 | 0.03 | 0.03 | 0.03 | 0.06 | 0.06 |
| $H_2O$ [b] | Range | 2.36–2.48 | 2.29–2.47 | 2.51–2.54 | 2.32–2.45 | 2.47–2.48 | 2.47–2.47 |
| | Average | 2.43 | 2.38 | 2.53 | 2.38 | 2.48 | 2.47 |
| Total | Range | 98.61–99.43 | 97.72–99.65 | 97.67–98.11 | 98.92–99.40 | 99.30–99.58 | 98.49–98.58 |
| | Average | 98.91 | 98.65 | 97.89 | 99.09 | 99.44 | 98.53 |

**Table 2.** *Cont.*

| Sample | | 94-5 | 94-6 | 94-8 | 127-2 | 127-4 | 127-9 |
|---|---|---|---|---|---|---|---|
| **Oxides (wt.%)** | | | | | | | |
| | | **Atomic Proportions Based on the Si = 6 apfu** | | | | | |
| $Si^{4+}$ | | 6.000 | 6.000 | 6.000 | 6.000 | 6.000 | 6.000 |
| $Ti^{4+}$ | Range | bdl | bdl–0.002 | bdl–0.003 | bdl–0.003 | bdl–0.002 | bdl–0.002 |
| | Average | | 0.001 | 0.001 | 0.001 | 0.001 | 0.001 |
| $Al^{3+}$ | Range | 1.823–1.929 | 1.755–1.923 | 1.693–1.744 | 1.772–1.908 | 1.694–1.716 | 1.824–1.827 |
| | Average | 1.861 | 1.842 | 1.732 | 1.851 | 1.705 | 1.831 |
| $V^{3+}$ | Range | bdl–0.001 | bdl–0.004 | 0.002–0.004 | bdl–0.003 | bdl–0.006 | bdl–0.001 |
| | Average | bdl | 0.001 | 0.003 | 0.001 | 0.003 | 0.001 |
| $Cr^{3+}$ | Range | 0.001–0.016 | 0.003–0.012 | 0.010–0.016 | 0.002–0.018 | 0.068–0.069 | 0.006–0.008 |
| | Average | 0.006 | 0.007 | 0.013 | 0.008 | 0.069 | 0.007 |
| $Be^{2+}$ | Range | 2.873–2.948 | 2.844–3.022 | 2.979–3.050 | 2.780–2.888 | 2.938–2.979 | 2.996–3.007 |
| | Average | 2.903 | 2.940 | 3.004 | 2.828 | 2.959 | 3.002 |
| $Mg^{2+}$ | Range | 0.226–0.295 | 0.231–0.336 | 0.312–0.318 | 0.222–0.300 | 0.317–0.337 | 0.280–0.294 |
| | Average | 0.269 | 0.283 | 0.316 | 0.258 | 0.327 | 0.287 |
| $Ca^{2+}$ | Range | 0.003–0.005 | 0.001–0.006 | 0.003–0.005 | 0.002–0.006 | 0.003–0.004 | 0.001–0.003 |
| | Average | 0.004 | 0.004 | 0.004 | 0.004 | 0.004 | 0.002 |
| $Fe^{2+}$ | Range | 0.029–0.041 | 0.033–0.051 | 0.040–0.048 | 0.031–0.054 | 0.054–0.057 | 0.034–0.039 |
| | Average | 0.036 | 0.041 | 0.045 | 0.039 | 0.056 | 0.037 |
| $Na^+$ | Range | 0.246–0.314 | 0.219–0.311 | 0.334–0.347 | 0.232–0.292 | 0.309–0.313 | 0.304–0.306 |
| | Average | 0.286 | 0.262 | 0.343 | 0.259 | 0.311 | 0.305 |
| $K^+$ | Range | 0.004–0.006 | 0.003–0.008 | 0.007–0.011 | 0.005–0.022 | 0.007–0.008 | 0.005–0.005 |
| | Average | 0.005 | 0.005 | 0.008 | 0.010 | 0.008 | 0.005 |
| $Li^+$ | Range | 0.019–0.019 | 0.019–0.030 | 0.019–0.026 | 0.015–0.026 | 0.015–0.019 | 0.019–0.049 |
| | Average | 0.019 | 0.026 | 0.022 | 0.019 | 0.017 | 0.034 |
| $Cs^+$ | Range | bdl–0.002 | 0.001–0.002 | 0.001–0.001 | 0.001–0.001 | 0.002–0.002 | 0.002–0.003 |
| | Average | 0.001 | 0.001 | 0.001 | 0.001 | 0.002 | 0.003 |
| $^cA^+$ | Range | 0.270–0.338 | 0.251–0.343 | 0.369–0.382 | 0.260–0.327 | 0.337–0.338 | 0.333–0.360 |
| | Average | 0.311 | 0.294 | 0.375 | 0.289 | 0.338 | 0.347 |

Compositions of emeralds were recalculated on the basis of Si = 6. bdl = below detection limit. apfu = atoms per formula units; $^cA^+$ = atoms of Channel monovalent alkali cations. [a] Oxides recalculated from LA-ICP-MS data. [b] Calculated using $H_2O$ = 0.5401 ln $Na_2O$ + 2.1867 [44].

LA-ICP-MS test results (Table 3) show that Indian emeralds tended to have high concentrations of alkali metal ions, ranging from 10,503 to 16,964 ppmw (avg. 13,942 ppmw); Na contents ranged from 9690 to 15,875 ppmw (avg. 12,950 ppmw); Cs contents ranged from 222 to 602 ppmw (avg. 377 ppmw); and Li contents ranged from 229 to 507 ppmw (avg. 340 ppmw). Chromophore concentrations of Cr, V, and Fe are 106–6310 ppmw (avg. 1266 ppmw), 37–122 ppmw (avg. 90 ppmw), and 2451–4153 ppmw (avg. 3468 ppmw), respectively; the concentrations of Cr were greater than V, and the Cr/V ratio ranged from 2.6 to 56. In addition, the emerald crystals also contain a certain amount of Sc (17–106 ppmw), Zn (37–108 ppmw), Ga (13.9–24.6 ppmw), Rb (30–83 ppmw), and other elements.

**Table 3.** Chemical composition (average) of Indian emeralds, obtained by LA-ICP-MS (in ppmw).

| Sample | | 127-9 | 94-8 | 127-4 | 94-5 | 94-6 | 127-2 | Detection Limit |
|---|---|---|---|---|---|---|---|---|
| Color | | Yellowish Green | Green | Bluish Green | Core-Rim | Core-Rim | Core-Rim | |
| **Element** | | | | | | | | |
| Li | Range | | 274–311 | 229–277 | 289–416 | 239–401 | 335–507 | 6.2 |
| | Average | 279 | 295 | 253 | 353 | 316 | 434 | |
| Be | Range | | 58,955–61,465 | 55,442–60,513 | 56,572–60,218 | 56,865–62,767 | 56,852–58,905 | 22 |
| | Average | 55,948 | 60,293 | 57,978 | 58,981 | 59,657 | 57,751 | |
| B | Range | | 7.65–19.21 | 9.66–29.58 | 1.69–10.75 | bdl-13.38 | 2.49–7.15 | 8.5 |
| | Average | 4.39 | 11.5 | 19.62 | 5.99 | 6.21 | 4.58 | |
| Na | Range | | 12,746–14,797 | 14,696–15,875 | 11,573–13,213 | 10,474–14,908 | 9690–13,707 | 355 |
| | Average | 14,007 | 13,504 | 15,286 | 12,282 | 13,067 | 11,732 | |
| Mg | Range | | 13,738–14,453 | 13,406–14,389 | 9570–14,466 | 9832–14,937 | 9675–13,277 | 52 |
| | Average | 12,900 | 14,189 | 13,898 | 12,311 | 12,476 | 11,327 | |
| Al | Range | | 53,693–58,603 | 58,129–59,099 | 60,300–70,313 | 59,400–63,833 | 54,455–62,002 | 52 |
| | Average | 58,432 | 55,692 | 58,614 | 64,597 | 61,741 | 59,398 | |
| P | Range | | bdl-69 | bdl-100 | 39–85 | bdl-81 | bdl-137 | 84 |
| | Average | 18 | 39 | 50 | 60 | 45 | 47 | |
| K | Range | | 169–305 | 269–315 | 173–241 | 108–325 | bdl-402 | 173 |
| | Average | 260 | 250 | 292 | 198 | 243 | 151 | |
| Ca | Range | | 750–1012 | 197–1082 | 315–1272 | bdl-1479 | bdl-3087 | 1020 |
| | Average | 1149 | 860 | 640 | 653 | 921 | 1403 | |
| Sc | Range | | 85–106 | 102–104 | 17–67 | 20–99 | 24–59 | 7.7 |
| | Average | 59 | 95 | 103 | 49 | 59 | 39 | |
| Ti | Range | | 0.1–13.8 | 1.1–29.8 | 0.2–13.9 | 5–32.2 | 5.6–24.4 | 18.3 |
| | Average | 15.2 | 5.1 | 15.5 | 8.9 | 18.1 | 15 | |
| V | Range | | 106–122 | 111–111 | 37–105 | 48–120 | 41–102 | 2.1 |
| | Average | 106 | 114 | 111 | 82 | 87 | 74 | |
| Cr | Range | | 1242–1273 | 5937–6310 | 106–1332 | 162–1192 | 362–1364 | 7.3 |
| | Average | 362 | 1259 | 6124 | 813 | 484 | 719 | |
| Mn | Range | | 10–16 | 14–22 | 8–15 | 7–16 | 2–74 | 4.4 |
| | Average | 15 | 13 | 18 | 12 | 11 | 27 | |
| Fe | Range | | 3848–4073 | 3874–3946 | 2759–3765 | 2451–4153 | 2627–3859 | 311 |
| | Average | 3419 | 3938 | 3910 | 3304 | 3394 | 3208 | |
| Co | Range | | 1.39–1.75 | 1.14–3.13 | 0.89–2.34 | bdl-3.48 | 0.64–2.82 | 1.9 |
| | Average | 1.52 | 1.5 | 2.14 | 1.78 | 1.64 | 1.70 | |
| Zn | Range | | 53–62 | 44–57 | 37–81 | 38–108 | 61–97 | 2.7 |
| | Average | 48 | 56 | 51 | 57 | 68 | 78 | |
| Ga | Range | | 21–23 | 17–20 | 14–20 | 15–20 | 17–24 | 1.6 |
| | Average | 25 | 22 | 19 | 16 | 19 | 20 | |
| Rb | Range | | 55–73 | 61–63 | 41–83 | 30–79 | 37–80 | 3 |
| | Average | 41 | 63 | 62 | 55 | 54 | 56 | |
| Cs | Range | | 365–408 | 403–479 | 222–459 | 249–602 | 281–572 | 1.9 |
| | Average | 423 | 383 | 441 | 314 | 388 | 362 | |
| Th | Range | | bdl | bdl | bdl | bdl | bdl | 0.04 |
| | Average | 0.058 | | | | | | |
| U | Range | | bdl | bdl | bdl | bdl-0.066 | bdl-0.024 | 0.01 |
| | Average | bdl | | | | bdl | bdl | |

bdl = below detection limit.

## 5. Discussion

### 5.1. Optical Properties of Indian Emeralds

The colors of Indian emerald include green or green with yellow tone, and bluish green, and the crystals are usually large and complete, but the transparency is poor. Hexagonal color zoning was prominent, often showing a greenish-white core and a medium-green

rim (Figure 4). According to the DiamondView™ test, the abnormal spotted blue-white fluorescence of Indian emerald was caused by the luminescence of albite. Moreover, the yellowish-brown fluorescence at crystal cracks was related to cedarwood oil, which was also confirmed in the infrared spectrum (the absorption peaks of 2873, 2875, 2979, and 2989 cm$^{-1}$) (Figure 12). As Indian emerald has a high content of Fe (2451–4153 ppmw), all samples show dark green under the Chelsea filter. This feature can help distinguish between Indian and Colombian emeralds, as well as other various emerald occurrences (those emeralds that appear light pink to dark red in their entirety under the Chelsea filter) to a limited extent.

### 5.2. Micro-Inclusions and Geological Environment

Mineral and fluid inclusions of Indian emerald are abundant. Talc, aragonite, albite, and phlogopite were identified by Raman analysis. An Indian emerald with long, columnar, surface-reaching quartz inclusion was identified by EMPA analysis (Figure 5b). Quartz inclusions within emerald have been found in emeralds from Brazil, Colombia, Russia, Madagascar, and Tanzania [45]. The country rocks of the Rajasthan emerald deposit consist of a set of Precambrian phyllite, with biotite and muscovite schists, a small amount of feldspar, talc schist, and vermiculite schist; they are intruded by quartz veins, granitic pegmatites, and tourmaline-bearing granitic rocks [15]. The minerals found in this study are closely related to the surrounding rock type of Rajasthan emerald deposits. Rajasthan emeralds typically occur in the phlogopite zone (near the pegmatite) or the talc-actinolite (-tremolite) zone (near the ultramafics) [15]. The solid inclusions and fluid inclusions within emerald crystals can reflect the surrounding geological environment during mineral crystallization [46–49].

### 5.3. Spectroscopy

Based on the UV-vis-NIR absorption spectrum, the most obvious distinction of Indian emeralds is the significantly increased intensity of the Fe-related 851 nm absorption band and the $Fe^{3+}$ absorption band at 369 nm due to their much higher Fe content. This feature has been used to easily separate Indian emeralds from those of Colombia, Brazil, China (Yunnan), and Afghanistan (Panjshir), [5,22,50–55]. However, the UV-Vis-NIR spectra of Indian emeralds are similar to those of Zambia and Nigeria [38]. However, Nigeria and Zambian emeralds can be distinguished from Indian emeralds by their chemical composition [38].

The absorption strength of type II $H_2O$ increases with the increase of alkali metal content, which indicates that emeralds from India may have a higher content of alkali metal ions (10,503 to 16,964 ppmw; avg. 13,942 ppmw) compared with emeralds from alkali poor structure water type.

### 5.4. Trace-Element Chemistry

There are few published trace element data of Indian emeralds, and usually only a few groups of data are used for origin comparison [1]. In this study, 20 spots on 13 Indian emeralds were analyzed via LA-ICP-MS. When distinguishing geographic origin, plots of trace element are used to detect possible correlation. Therefore, it is necessary to add LA-ICP-MS data from other major emerald deposits in the world, such as Rajghar from India, Swat Valley from Pakistan, Ethiopia, Russia, Madagascar, Mozambique, Tanzania, Zambia, Zimbabwe, Panjshir from Afghanistan, Austria, Brazil, Colombia, and Egypt [1,4,53,56]. To explore the discrimination further, a series of binary diagrams were constructed (Figures 14–16).

In the Li versus Cs binary diagram (Figure 14), the Cs content in Brazil (907–980 ppmw), Zimbabwe (739–915 ppmw), and Ethiopia (650–675 ppmw) [1] emeralds is relatively higher than the Cs in the Indian emeralds from this study (222 to 602 ppmw), although they show a wide range of variation. Indian emeralds contain moderate amounts of Li (229 to 507 ppmw).

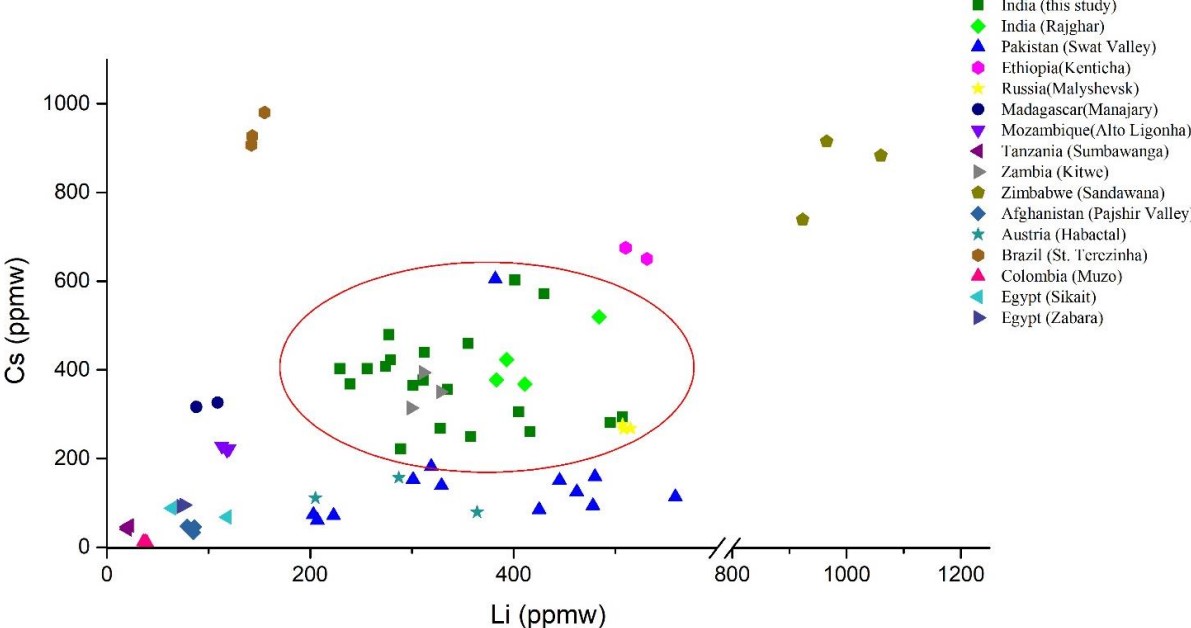

**Figure 14.** Plot of lithium (Li) versus cesium (Cs) concentrations from LA-ICP-MS analyses. Data are expressed in ppmw. Other sources are from [1,4,53,56].

The log plot of Li versus Sc proved useful in separating India from Swat Valley of Pakistan (Figure 15). The Sc content of Indian emerald ranges from 17 to 106 ppmw. In contrast, Swat emeralds contain relatively high scandium (avg. 633 ppmw); there is a clear boundary of Sc content between Indian and Swat emeralds. Apart from that, Swat emeralds have the highest Mg concentrations (avg. 34,263 ppmw) among world-wide sources [4].

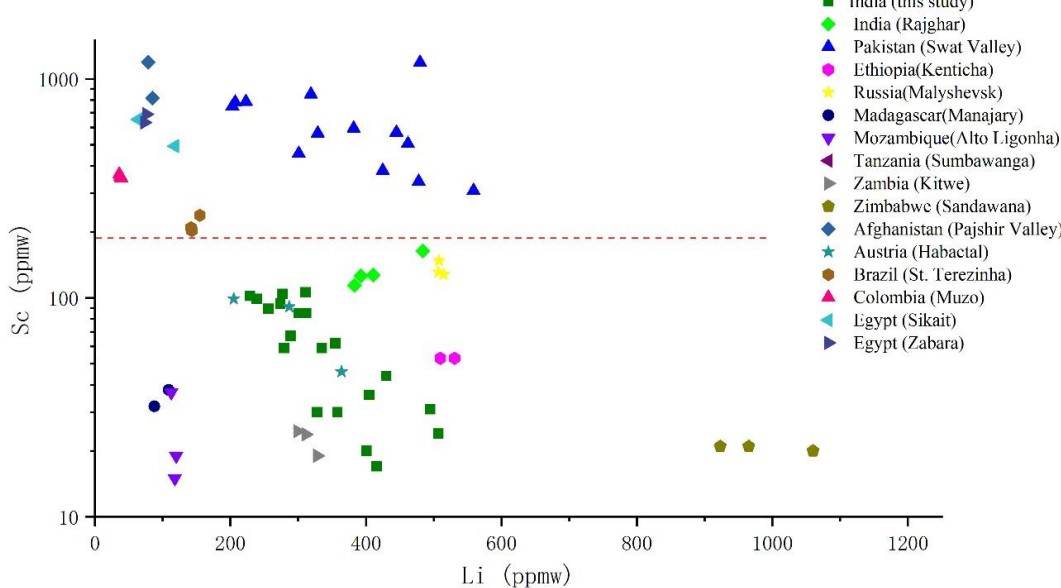

**Figure 15.** A log plot of lithium (Li) versus scandium (Sc) concentrations in emeralds from sixteen deposits. Data are expressed in ppmw. Sources of data are the same as Figure 14.

In this study, we have collected several public data from Rajghar in India [1]. The V content of Rajghar emerald range from 323 to 343 ppmw, and the V content in the samples of this study are relatively lower than Rajghar (Figure 16). The Cr/V ratio of Indian emerald is mainly between 2.6–13.8, and the bluish green sample showed a high

Cr/V ratio (53.5 and 56.7). It can be seen that Cr has a greater effect on color than V in Indian emeralds.

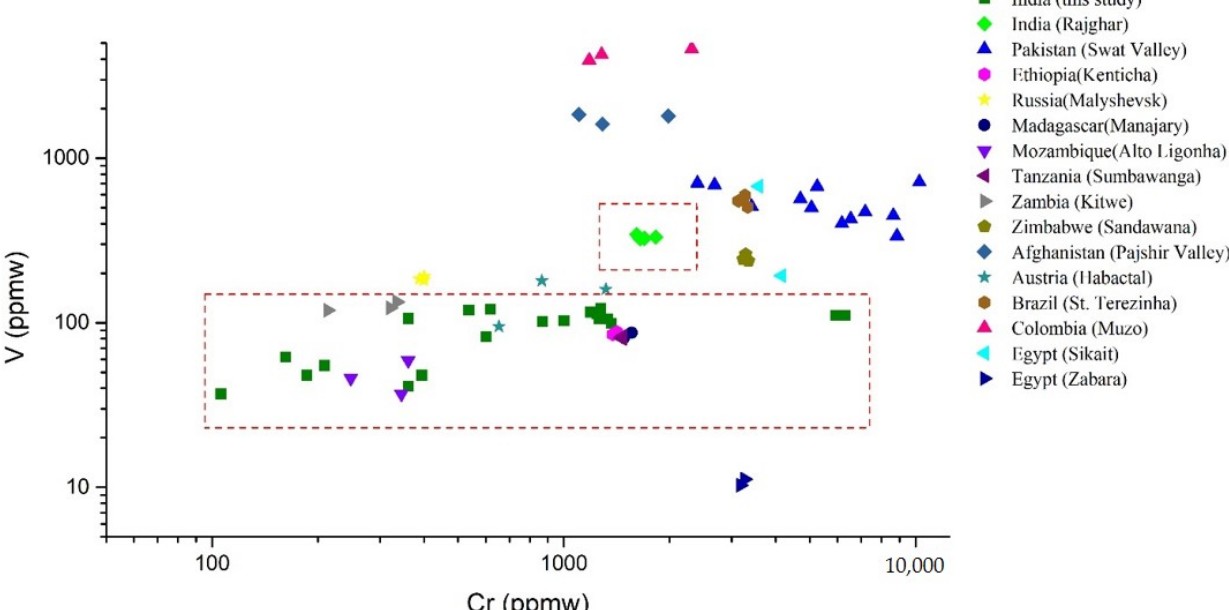

**Figure 16.** A log-log plot of chromium (Cr) versus vanadium (V) concentrations in emeralds from sixteen deposits. Data are expressed in ppmw. Sources of data are the same as in Figure 14.

Compositional changes of trace elements for sample 94-6 are shown in Figure 17. With the change of crystal color, the content of trace elements shows a variability. The content of most elements is relatively stable in the core, and changes excessively from the greenish-white core to the medium-green rim, indicating that the growth environment of emerald at the initial stage of crystallization is relatively stable. The contents of Fe (2451 to 4153 ppmw), V (48 to 120 ppmw), Cr (162 to 1192 ppmw), Sc (20 to 99 ppmw), Cs (249 to 440 ppmw), and Rb (30 to 79 ppmw) in trace elements showed a significant upward trend from the core to the rim. From the core to the edge, the contents of Ga (15 to 20 ppmw) fluctuate without obvious change.

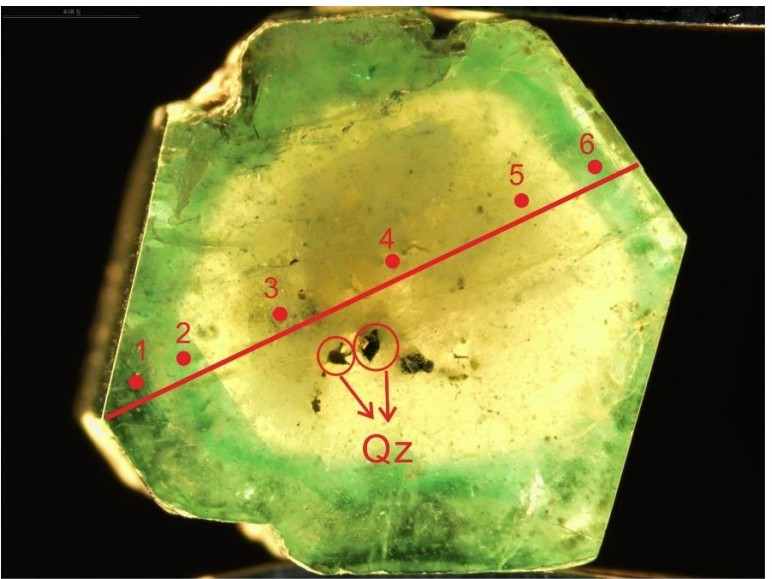

**Figure 17.** *Cont.*

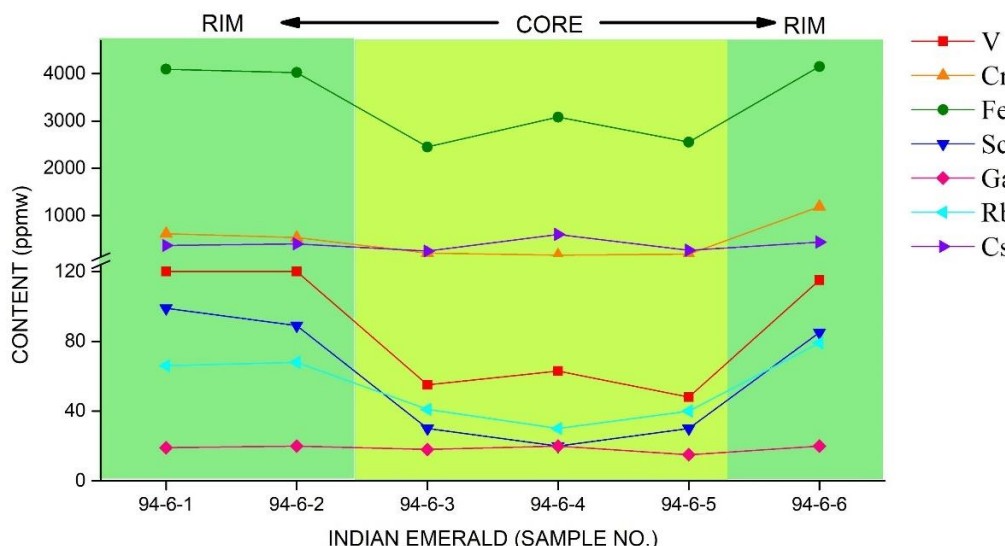

**Figure 17.** Representative photomicrograph of color zoned emerald crystal 94-6 with a greenish-white core and a medium-green rim. The red circles indicate the positions of analysis points corresponding to analyses 94-6-1 to 94-6-6. LA-ICP-MS data for selected trace element analysis from 94-6-1 to 94-6-6. Plots of V, Cr, Fe, Sc, Ga, Rb, and Cs are used to highlight compositional changes, with uneven ordinate scales for clearer display.

## 6. Conclusions

This work provides new data on Indian emeralds, including the composition of multiphase inclusions by Raman. The detailed major elements (by EMPA) and trace elements (by LA-ICP-MS) of Indian emerald were determined, which improved more data support for the origin identification of emeralds.

Indian emeralds showed some special optical properties, such as being inert to long- and short-wave UV radiation and show dark green under the Chelsea filter. We have found that the characteristics of mineral inclusions (phlogopite, quartz, talc, aragonite, and albite) and fluid inclusions correspond to the geological characteristics of Indian Rajasthan emerald deposits. This is the first report on the chemical composition of hexagonal multiphase inclusions of Indian emeralds, which include gaseous $CO_2$ (or $CO_2 + CH_4$), aqueous solution (or liquid mixtures of $H_2O + CO_2$), siderite, dolomite, phlogopite, and magnesite. The characteristic absorption peaks related to iron in UV-Vis-NIR spectra confirm that Indian emerald belongs to schist-hosted. The higher alkali metal content shows an obvious type II $H_2O$ absorption peak in the infrared spectrum.

Chemical composition tests show that Indian emerald contains a wide range of chromium (106–6310 ppmw), high alkali metals (avg. 13,942 ppmw), moderate iron (avg. 3468 ppmw), and low vanadium (avg. 90 ppmw). The logarithmic map, combined with EMPA data, can distinguish Indian emeralds from a number of emeralds elsewhere that are clearly distinguishable. However, in order to determine the origin accurately, we must carefully use multiple pieces of evidence to arrive at a conclusion.

**Author Contributions:** Conceptualization, L.-J.Q. and X.-Y.Y.; methodology, L.-J.Q.; software, L.-J.Q.; validation, L.-J.Q., H.-S.G. and X.-Y.Y.; formal analysis, L.-J.Q., H.-S.G. and X.-Y.Y.; investigation, L.-J.Q. and X.-Y.Y.; resources, H.-S.G.; data curation, L.-J.Q.; writing—original draft preparation, L.-J.Q.; writing—review and editing, L.-J.Q. and X.-Y.Y.; visualization, L.-J.Q. and X.-Y.Y.; funding acquisition, X.-Y.Y. All authors have read and agreed to the published version of the manuscript.

**Funding:** This research was funded by China Geological Survey Project "Geology of Mineral Resources in China" (grant number DD20190379-88) to Xiao-Yan Yu.

**Data Availability Statement:** All data generated or used during the study appear in the submitted article.

**Acknowledgments:** We owe thanks to Zhen-Yu Chen from CAGS for assistance in EMPA analyses and express our gratitude to Jian-ming Li for LA-ICP-MS testing support. The authors are grateful to Zheng-Yu Long and Yu-Yu Zheng for constructive comments, and Jia-Xin Wan in experimental assistance.

**Conflicts of Interest:** The authors declare no conflict of interest.

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
