# Peer review of "Fluid Inclusion and Chemical Composition Characteristics of Emeralds from Rajasthan Area, India"

_minerals, doi:10.3390/min12050641_

Round 1

Reviewer 1 Report

line 108 How was the research material obtained? Did the study authors take samples? Did they take samples in person or were they purchased?

line 362 Please specify localities that show a change below the Chelsea filter.

line 358 Why is oil present in some samples? As the source of the sampling is not known.

The article is very well compiled, it brings new information. All that is needed is to define the method of sampling. Whether the authors of the article dug them in the field or bought them. It is important whether the source is trustworthy.

Reviewer 2 Report

This is a decent paper and the provided information and results are useful.

I have a number of comments about wording and about details.

Four more important comments ahead:

1) Fig 13 is mislabeled or misinterpreted. Type-I and II H2O bands are at much lower energy and the shown peaks are too pronounced to be overtones. I suggest these are Cr- and Fe-related defect bands. Please clarify this issue.

2) Why do you not show LA-ICP-MS of the fluid inclusions? What is the point of LA-ICP-MS if you only analyse the bulk - you could have used ICP-MS.

3) You use fluid interchangeably with liquid. All former fluid inclusion that you show are in the liquid-vapour coexistence field, hence the phase that you labeled as 'fluid', 'aqueous fluid' etc. is actually a liquid!

The gemscience and the ecomonic geologist communities are sloppy in using the term fluid but this is not acceptable: Fluid is supercritical by definition and wherever there is a gaseous phase there is no fluid but a liquid coexisting.

Use 'former fluid' where you refer to the bulk and 'liquid' where you refer to the present non-gaseous and non-solid phases in the inclusions.

4) The EPMA totals are not good. This is very likely result of the insufficient ZAF correction for Be. Use the LA-ICP-MS data of Be to correct the EPMA.

Detailed comments:

The authors collected emerald samples of Rajasthan area, India from the Chinese 
71 market to investigate their gemological, spectral and chemical characteristics.

-> How do you know that they are from Rajasthan? By word of the dealers, composition, other characteristics (which)?

They (THE EMERALD OCCURRENCES?)
result from the metasomatic reaction between the peg-
82 matites and their ultrabasic host-rocks (or Delhi gneisses as at Bubani), which
? and are usually hosted?

 is usually
 83 hosted in the contact between muscovite (± garnet ± tourmaline) pegmatites and lenticular 84 bodies of altered ultra-mafic rock located in the Delhi Group gneisses [10].

and the yellow-brown mineral aragonite was identified on some samples (Figure 3). 
-> Change to: ‘and yellow-brown aragonite….’
(pure aragonite is colourless…)

Table:
short columnar black mineral; white cloud solid inclusion; tubular and needle-like inclusions. 

-> I understand that it may be hard to indentify all inclusions but with the set of techniques that you applied it should be possible to characterize these inclusions - this would be quite interesting. If you have information about them, please add it to the paper.

Two black columnar mineral inclusions were observed 204 in the greenish-white core of sample 94-6 (Figure 5b), and the EMPA test results showed 205 that the inclusions were quartz, the content of SiO2 was 100.18 wt.%, the content of FeO 206 was 0.097 wt.%, the content of Cr2O3 was 0.041 wt.%, 
-> Ok, quartz does not contain that much Fe and Cr!
 Hence, Fe and Cr are hosted by a different phase (chromite?) - please reassess the EPMA by assigning the elements to the proper phases. Use your Raman etc. data to check if the inclusion is really chromite or rather a different mineral.

 Indian emeralds contain black mineral and disorderly dis-209 tributed short needle inclusions (Figure 5e)

-> Change to : ‘… contain A black opaque mineral and un-oriented needle-shaped inclusions (of what appearance: white, back, coloured…?)

What are these inclusions? They should be characterized.

A group 211 of tubular inclusions oriented parallel to the c-axis (Figure 5g) and needle-like inclusions 212 (Figure 5h) were also visible on emerald interior. 
-> Good, but what are they? I think you refer at least in part to the fluid inclusions - if this is true, you should clarify this point here already. Otherwise the reader receives the impression that much/most of the inclusions are not characterized.

-> IN the interior!!!

Figure 8 etc: You use the term ‘fluid’ but the inclusions are clearly in the vapour-liquid coexistence field. Hence, the non gaseous phase is no longer a fluid, but a liquid. Specify which liquid it is (water, brine,…). 
Both, the gas and the liquid WERE a fluid when they were entrapped, but it is important to keep the terminology clear (even if some economic geologists etc tend not to do this ).

‘aqueous fluid’ : Water! (see above). 
Fluid is NOT liquid. Your inclusion are in the v-l coexistence field and not above the critical point!
Please make this clear through proper wording.

Fe/Cr absorption bands: Good, but clarify whether they are from the emerald host or from the Fe-Cr rich inclusions!

Figure 12: Why does the absorbance show negative dips!? It appears that the background was oversubtracted? Please redo the background subtraction such that no bands of ‘negative absorbance’ appear. 

Figure 13: Either the X-axis scale is wrong or you show smth else - there are ‘H2O’ bands at that high energy and your references state ‘type I and II’ bands at much lower energies!
This needs to be corrected!

These bands may be transitions from Fe and Cr?!

They are too strong for overtones of ‘type I and II H2O bands.

Table 3: Convert the major elements to wt% oxide to allow comparison to the EPMA.
The sums of the EPMA are quite low but this may be owed to insufficient matrix correction for Be - hence, you could correct Be by using the LA-ICP-MS data (otherwise, and besides for the traces - why did you bother collecting LA-ICP-MS data?).

Same for Li, Na…

Did you collect LA-ICP-MS on individual fluid inclusions? Why not?

Round 2

Reviewer 1 Report

Comments were incorporated and questions were answered.

Reviewer 2 Report

all comments are addressed. the paper is acceptable.